# Single-Cell RNA Sequencing and Microarray Analysis Reveal the Role of Lipid-Metabolism-Related Genes and Cellular Immune Infiltration in Pre-Eclampsia and Identify Novel Biomarkers for Pre-Eclampsia

**DOI:** 10.3390/biomedicines11082328

**Published:** 2023-08-21

**Authors:** Yujie Liu, Borui Xu, Cuifang Fan

**Affiliations:** 1Department of Obstetrics and Gynecology, Renmin Hospital of Wuhan University, Wuhan 430060, China; 2021283020238@whu.edu.cn; 2Department of Pancreato-Biliary Surgery, The First Affiliated Hospital, Sun Yat-sen University, Guangzhou 510080, China; xubr@mail2.sysu.edu.cn

**Keywords:** pre-eclampsia, lipid metabolism, single-cell analysis, immune infiltration, novel biomarkers

## Abstract

Pre-eclampsia (PE) is a gestational hypertensive disorder that is characterized by hypertension and proteinuria, typically occurring after 20 weeks of gestation. Despite its global impact on pregnant women, the precise pathogenic mechanisms of PE remain unclear. Dysregulated lipid metabolism and immune cell infiltration contribute to PE development. Our study aimed to identify lipid-metabolism-related genes (LMRG-PEs) and investigate their association with immune infiltration. We utilized the “Seurat” R package for data quality control, cell clustering, and marker gene identification. The “SingleR” package enabled the matching of marker genes to specific cell types. Pseudotemporal ordering analysis was conducted using the “Monocle” package. Weighted correlation network analysis (WGCNA), gene set variation analysis (GSVA), and gene set enrichment analysis (GSEA) approaches were employed to explore lipid-metabolism-related genes, while potential targeted drugs were predicted using the drug–gene interaction database (DGIdb). Hub gene expression was validated through RT–qPCR. By analyzing single-cell RNA sequencing data, we identified and classified 20 cell clusters into 5 distinct types. Differential gene expression analysis revealed 186 DEGs. WGCNA identified 9 critical modules and 265 genes significantly associated with PE diagnosis, emphasizing the importance of the core genes *PLA2G7* and *PTGS2*. RT–qPCR confirmed the significantly decreased expression of *PLA2G7* and *PTGS2* in PE patient tissues. These findings offer valuable insights into the molecular mechanisms of PE, particularly those involving lipid metabolism and immune infiltration. The identified hub genes have potential as therapeutic targets and biomarkers for future research and clinical applications.

## 1. Introduction

PE is a placenta-dependent disorder that is defined as the presence of new-onset hypertension, proteinuria, and multiple organ dysfunction (mainly that of the kidney, brain, and liver) occurring after 20 weeks of gestation [1,2]. It carries an estimated risk of 2% to 8% during pregnancy, making it a prevalent condition associated with significant maternal and neonatal morbidity and mortality [2]. PE is widely recognized as a complex interaction among multiple genetic components [3,4], angiogenic factors [5], metabolic pathways [6], immunologic aberrations [7], and certain acquired risk factors [8,9,10]. PE is hard to predict since it is usually symptomless in the early phase. Seizures, dyspnea, epigastric pain, and profound placental abruption are indicative symptoms of a critical terminal phase [11,12,13]. The diagnosis and prediction of PE was established by assessing the presence of de novo hypertension and proteinuria, as well as the identification of placental or maternal-derived circulating biomarkers [14].

At present, the clinical outcomes of PE are improved by implementing preventive measures, a timely diagnosis, vigilant monitoring, and the appropriate administration of medications, as deemed necessary. PE continues to be a significant contributor to morbidity and mortality in both developing and developed nations, despite the routine utilization of aspirin [15,16,17] and other promising therapeutic interventions. In PE, levels of soluble fms-like tyrosine kinase-1 (sFlt-1) increase, whilst placental growth factor (PlGF) levels decrease. sFlt-1, acting as an antagonist to PlGF, can lead to vascular issues and potentially early-onset PE. An elevated sFlt-1 to PlGF ratio is associated with this risk, and effective screening through this ratio has been implemented in the second and third trimesters [18]. Once diagnosed, PE necessitates delivery of the placenta and fetus as the sole definitive therapeutic approach, rather than pursuing expectant management [2]. Palliative treatment primarily encompasses the administration of antihypertensive therapy to prevent maternal intracranial hemorrhage, in addition to the use of magnesium sulfate (MgSO4) as an anticonvulsant therapy [14,19]. Nonsteroidal anti-inflammatory drug (NSAID) medications should continue to be used preferentially over opioid analgesics for controlling postpartum pain [20]. Research has shown that the administration of low-dose aspirin from early pregnancy until 36 weeks can effectively lower the likelihood of developing PE [17]. Since PE syndrome is hard to predict, we are committed to performing bioinformatics for screening to identify PE-related biomarkers and therapeutic targets, as well as help develop effective diagnostic strategies and appropriate treatment and predict the prognosis of patients [21,22].

Although PE significantly contributes to maternal and fetal morbidity and mortality, the underlying pathophysiological mechanisms remain poorly understood. Consequently, this knowledge gap presents a critical opportunity for the advancement of innovative diagnostic approaches and therapeutic interventions. From a pathophysiological perspective, the primary cause of PE lies in placental dysfunction rather than fetal factors [14]. This placental defect is most likely associated with abnormal lipid metabolism [23] and partial breakdown of maternal–fetal immune tolerance [24,25]. Fatty acid oxidation (FAO) disorders and dyslipidemia are involved in the pathogenesis of PE through oxidative stress and inflammatory endothelial cell injury [23,26,27]. The deposition of lipids within the endothelium leads to heightened atherosclerosis, culminating in the development of abnormally constricted spiral arterioles in PE and subsequent impairment of placental perfusion [22,28]. Numerous studies [29,30] have consistently demonstrated that the pathogenesis of PE is influenced by an autoantibody known as AT1-AA, which specifically targets the angiotensin II type 1 receptor. Prior investigations have shown that LXA4, an endogenous lipid mediator known for its anti-inflammatory and pro-resolution properties, exerts a suppressive effect on the production of AT1-AA through the activation of caspase-1 [31]. This study offers valuable insights into the molecular mechanisms underlying PE, unveiling novel therapeutic targets and identifying potential biomarkers. Additionally, it serves as a reference for future research and highlights the utility of bioinformatics analysis in investigating PE pathologies in humans.

In view of the importance of placental lipid metabolism and immune cell infiltration in the etiology of PE [32,33], this study aimed to investigate PE pathogenesis by identifying the pivotal LMRG-PE module (lipid-metabolism-related genes) and provide insights for future research. Utilizing single-cell RNA sequencing data, five distinct cell types in PE were identified, including monocytes and NK cells, suggesting their potential involvement in PE development. Weighted gene co-expression network analysis (WGCNA) identified two hub genes associated with PE. ROC curve analysis revealed *PLA2G7* and *PTGS2* as potential biomarkers for PE diagnosis. GSEA and GSVA indicated their associations with lipid metabolism, immune responses, and cellular processes, supporting their roles in PE pathogenesis. This study contributes to understanding the molecular mechanisms of PE and identifies potential diagnostic biomarkers and therapeutic targets.

## 2. Materials and Methods

### 2.1. Patients and Sample Collection

Sixteen placental tissue samples were collected from patients with PE and matched healthy control individuals from the Obstetrics and Gynecology Department, Renmin Hospital of Wuhan University (Wuhan, China) between July 2022 and October 2022 (Appendix A). We followed stringent inclusion criteria when procuring samples from patients with severe pre-eclampsia (Appendix A). All collected samples originated from patients diagnosed with early-onset pre-eclampsia with severe complications. The 16 cases obtained represent the maximum sample volume that we could amass within the specified time frame. We enhanced the robustness of our study by employing four sampling points and acquiring multiple samples from disparate locations (Appendix A). Informed consent was obtained from each participant prior to the study for the intended use of placental tissue samples. Placental tissues were collected from women undergoing cesarean section in the late stages of pregnancy. The placental specimens were rinsed with sterile PBS and subsequently preserved in liquid nitrogen for further analysis [34]. The collection of human samples was authorized by the Ethical Review Board of Renmin Hospital, Wuhan University (WDRY2021-K177) and conducted in accordance with the principles of the Helsinki Declaration.

### 2.2. Data Recruitment and Processing

The microarray-based RNA expression data corresponding to GSE48424 were retrieved from the GEO database (https://www.ncbi.nlm.nih.gov/geo/, accessed on 14 June 2023). This consisted of 19 PE patient samples and 19 control samples. The dataset was generated using the GPL6480 ([PrimeView] Affymetrix Human Gene Expression Array) platform. Additionally, the GSE192693 dataset contained single-cell RNA sequencing data from six PE samples and four normal samples based on 10× Genomics. For comprehensive processing of the scRNA-seq data, the following sequential steps were taken. The initial step of the analysis involved preprocessing the scRNA-seq data using the “Seurat” package. This preprocessing encompassed several procedures, including assessing the proportion of mitochondrial genes using the PercentageFeatureSet function and conducting correlation analysis to explore relationships between sequencing depth, mitochondrial gene sequences, and total intracellular sequences [35].

To ensure robust analysis, gene filtering was applied, requiring a minimum expression in at least 5 cells. Furthermore, a stringent selection process was implemented to retain cells that met specific criteria. These criteria encompassed gene expression counts ranging from over 300 to under 5000, mitochondrial content below 10%, and a minimum UMI count of 1000 per cell. Upon completing the data filtering steps, normalization of the scRNA-seq data was carried out using the LogNormalize method, enhancing the accuracy of downstream analysis and interpretation.

For downstream analysis, the top 20 principal components (PCs) were subjected to Seurat’s Elbow plot program. Primary cell clusters were identified using Seurat’s Find Clusters tool with a resolution of 1.2. These clusters were subsequently visualized using two-dimensional t-SNE or UMAP plots. Each cell was classified into a recognized biological cell type using conventional markers established in prior studies. From the MSigDB database [36], we obtained a total of 418 LMRGs.

### 2.3. Identification of DEGs and Construction of Weighted Gene Co-Expression Networks

The limma package in R was used to identify DEGs with adjusted *p <* 0.05 and |Log2 fold change| > 1. Heatmaps and volcano plots were generated utilizing the Pheatmap and ggplot2 packages in the data analysis process. The WGCNA package [37] was used to analyze the gene co-expression network of the GSE48424 dataset. The samples were subjected to clustering analysis, and outliers were subsequently removed from the dataset. A scale-free network was built with a soft threshold of *β* = 2. Module-gene correlations were calculated using WGCNA. Modules highly correlated with PE were selected. A total of 451 PE-related genes were obtained by intersecting DEGs and key module genes. They were cross-referenced with LMRG using “VennDiagram”. Finally, 11 LMRG-PE expression profiles were obtained.

### 2.4. KEGG and GO Enrichment Analysis

LMRG-PE was functionally annotated using the R package “clusterProfiler” [38], containing Gene Ontology (GO) and Kyoto Encyclopedia of Genes and Genomes (KEGG) pathway analysis. GO terms comprised biological processes (BPs), cellular components (CCs), and molecular functions (MFs) [39] and were utilized for the identification of biological characteristics pertaining to genes and genomes across all organisms. Enrichment pathway analysis using KEGG was performed to identify pathways associated with the studied conditions. A statistically significant threshold was set at an adjusted *p* < 0.05.

### 2.5. ROC Curve Analysis and Expression Analysis

Using the GSE48424 dataset, we employed 19 PE samples and 19 control samples to generate receiver operating characteristic (ROC) curves. Using the “pROC” package, a robust tool for assessing the diagnostic performance of genes, the area under the ROC curve (AUC) was computed. Core genes with an AUC > 0.7 were identified as valuable biomarkers for disease diagnosis. Boxplots generated utilizing the R package “ggplot2” were employed to visualize the expression levels of these core genes.

### 2.6. Correlation Analysis between Core Genes and Infiltrated Immune Cells

Immune infiltration analysis was conducted utilizing the single-sample gene set enrichment analysis (ssGSEA) algorithm. Correlation analysis was conducted using the Spearman method to assess the relationship between essential genes and 28 immune cell types. The results were visually presented. Furthermore, to determine the association between core genes and distinct immune cell populations, correlation analysis was performed.

### 2.7. Prediction of Networks Mutually Regulated by miRNAs and TFs

The miRNet database [40] was employed to predict the upstream transcription factors (TFs) and miRNAs. The resulting predictions were visualized using Cytoscape software v.3.8.2 (U.S. National Institute of General Medical Sciences (NIGMS), Bethesda, MD, USA).

### 2.8. Laboratory Measurements

This study involved RNA isolation, followed by quantitative reverse transcription polymerase chain reaction (qRT–PCR) analysis. The results obtained from this study were validated using qRT–PCR after RNA was isolated from human placental tissue. Total RNA extraction was performed using RNAiso Plus (TRIzol) reagent, followed by analysis using a NanoDrop 2000 spectrophotometer. Based on the instructions provided by the manufacturer, the TSK301 reverse-transcription system kit was used for the RT reaction and quantitative real-time PCR. The qRT–PCR analysis employed SYBR Green qRT–PCR Master Mix. (all from Servicebio, Wuhan, China). A primer was designed and synthesized by Wuhan Servicebio Co., Ltd. An amplification procedure of 40 cycles was conducted at 95 °C for 10 s, 60 °C for 30 s, and 60 °C for 30 s of denaturation, annealing, and extension, respectively. To normalize the expression levels of the target genes, GAPDH was utilized as an internal reference. The following is a list of the primer sequences for each signature:

*PLA2G7*: TCAATGACAACTCCTGCAAACTG (sense primer);*PLA2G7*: TCCTCCTCTTGTTTCAGGGTTCT (antisense primer);*PTGS2*: GGGTTGCTGGTGGTAGGAATG (sense primer);*PTGS2*: CATAAAGCGTTTGCGGTACTCAT (antisense primer);*GAPDH*: GGAAGCTTGTCATCAATGGAAATC (sense primer);*GAPDH*: TGATGACCCTTTTGGCTCCC (antisense primer).

### 2.9. Statistics

In this study, all parametric analyses were performed using two-tailed tests, with a significance threshold set at *p* < 0.05. Statistical analysis was performed using R software v.4.1.3 (R Foundation for Statistical Computing, Vienna, Austria). Unless explicitly mentioned, group comparisons for categorical and continuous variables were respectively assessed using Fisher’s exact test and *t*-test assuming equal variances. ROC curve analysis and calculation of the AUC values were performed to assess the diagnostic accuracy of gene expression levels in predicting pre-eclampsia. Statistical significance was defined as *p <* 0.05, unless stated otherwise.

## 3. Results

### 3.1. scRNA-Seq Data Preprocessing and PCA

After data filtering, the scRNA sequencing dataset contained a total of 15,311 clean cells. Subsequent processing of the dataset using PCA, Harmony, and t-SNE techniques allowed us to examine the results. Following quality analysis of the PE scRNA-Seq data, we excluded zero low-quality cells. We observed a positive correlation between the number of detected genes and the total number of genes identified through sequencing.

Figure 1A displays a strong association between the number of UMIs and mRNAs, but no correlation was found between the number of UMIs/mRNAs and the content of mitochondrial genes. Violin plots before and after quality assurance are shown in Figure 1B,C. By performing variance analysis on 18,357 genes, we identified the top 2000 genes that exhibited the highest variation. Among them, the top 10 genes were *HBB*, *HBA2*, *HBA1*, *IGHV3−66*, *ALAS2*, *IGKV3−15*, *JCHAIN*, *CA1*, *STMN1*, and *MZB1* (Figure 1E).

To estimate the available dimensions, we employed principal component analysis (PCA) and found no significant distinction between PE cells. Although the top 2000 genes underwent PCA dimension reduction, PC_1 and PC_2 did not demonstrate significant discriminatory capability among cells from different PE samples (Figure 1D). The distribution of cells from the pre-eclampsia and control groups is shown in Figure 2A. Finally, we selected 20 principal components (PCs) based on evaluation (*p* < 0.05) for subsequent analyses (Figure 2B).

### 3.2. Cell-Type Annotation and Single-Cell Differentiation Trajectory Analysis

Nineteen clusters were assigned to known cell lineages based on marker genes, as reported in a previous study. Cell annotation was performed using the R package “SingleR”, which identified a total of five cell types (Figure 2C). Monocytes were annotated to clusters 1, 3, 4, 5, 6, 7, 11, 14, and 18. NK cells were annotated to clusters 0 and 9. T cells were annotated to clusters 2, 10, 15, 17, and 19. Platelets were annotated to clusters 8 and 12. HSC-G-CSF was annotated to clusters 13 and 16.

Subsequent trajectory analysis was conducted on the annotated cells. To explore the potential relationships between different cell types, we performed pseudotime trajectory analysis. The results revealed that these cells could be divided into three states with a common origin (Figure 2D). Cluster 1 was found at the beginning of the trajectory, while most of the cells were mainly concentrated in states 4 to 15 (Figure 2E).

### 3.3. Immune Characterization Analysis

To investigate the relative proportion of the 22 immune cell types, the CIBERSORT algorithm was implemented on a dataset comprising 16 PE samples and 16 normal control samples. Additionally, to gain insights into the immune microenvironment of PE, the ssGSEA technique was employed to analyze specific immune cell types within the PE patient cohort (Figure 3A). Furthermore, a visually informative heatmap was generated to depict the ssGSEA estimation of immune cell infiltration (Figure 3B). The levels of activated NK cells and M2 macrophages were significantly elevated (*p* < 0.05) in the PE group (Figure 3C). Conversely, the abundance of neutrophils was higher in the normal control group. These observations strongly suggest the potential involvement of immune cells in mitigating the incidence rate of PE.

### 3.4. Identification of DEGs and Construction of Co-Expression Networks

Our investigation yielded a comprehensive set of findings. A total of 186 differentially expressed genes (DEGs) were identified. Of these DEGs, 67 genes were significantly upregulated and 119 genes were significantly downregulated in the PE samples compared with the control samples (Figure 4A). The heatmap displayed in Figure 4B illustrates the top 10 upregulated and downregulated DEGs. The sample clustering tree, as depicted in Figure 5A, indicated the absence of any abnormal samples. Subsequently, through thorough calculations, we determined that the optimal soft-thresholding power was set at 2 (Figure 5B,C). Consequently, a distinct color was assigned to each module in GSE48424, resulting in a total of nine modules. (Figure 5C,D). Furthermore, Figure 5D illustrates the outcomes of the module–feature relationship analysis, indicating a robust correlation between PE and the green module (correlation coefficient = −0.48, *p* value = 0.04). Consequently, a total of 265 genes belonging to the green module were selected for subsequent investigation and analysis.

### 3.5. Identification of LMRG-PE and Functional Enrichment Analysis

Then, after taking the intersection of PE-related genes and LMRGs, we obtained 11 LMRG-PEs (Figure 6A). The heatmap shows the expression of 11 candidate genes ordered by adjusted *p*-value (Figure 6C). To explore the pathways of 11 LMRG-PEs, KEGG analysis was performed. As depicted in Figure 6B, these 11 LMRG-PEs (liver metabolism-related genes associated with pre-eclampsia) are primarily enriched in processes related to primary bile acid biosynthesis, fat digestion and absorption, and arachidonic acid metabolism. GO analysis was conducted to explore the functional characteristics of these 11 LMRG-PEs, with the corresponding GO terms shown in Figure 6D. LMRG-PEs were found to be primarily involved in lipid metabolism and biosynthetic processes. In MF analysis (Figure 6E), LMRG-PE was involved in the activities of lipid transporters, steroid hydroxylase, and transmembrane transporters.

### 3.6. Expression Analysis of Core Genes at the Single-Cell Level and Bulk RNA-seq Level

In the single-cell expression analysis of the 11 LMRG-PEs, our findings revealed that *PTGS2* was enriched in the control group (Appendix A). Furthermore, both *PTGS2* and *PLA2G7* showed enrichment in monocytes. To assess the expression levels of these 11 LMRG-PEs between PE and control samples, we examined the GSE48424 dataset. Interestingly, we observed significantly lower expression levels of two key genes (*PLA2G7* and *PTGS2*) in the PE samples, indicating the most prominent changes (Appendix A).

To evaluate the sensitivity and specificity of PE diagnosis, we performed ROC curve analysis, as illustrated in Appendix A. The diagnostic accuracy of *PLA2G7* and *PTGS2* in identifying PE was evident from their respective AUC values of 0.836 and 0.735, indicating high sensitivity and specificity.

### 3.7. Associations of Core Genes with Immune Cells

In this study, Spearman correlation analysis was conducted to investigate the potential association between *PLA2G7* and *PTGS2* and immune cell infiltration. During the correlation analysis, a positive relationship was observed between *PLA2G7* and natural killer T cells, myeloid-derived suppressor cells (MDSCs), effector memory CD8 T cells, and central memory CD4 T cells. Conversely, a negative correlation was found between *PLA2G7* and mast cells. (Figure 7A). *PTGS2* exhibited a significantly positive correlation with neutrophil cells, mast cells, macrophages, eosinophil cells, central memory CD4 T cells, CD56 bright and CD56 dim natural killer (NK) cells, and activated CD4 T cells (Figure 7A). Specifically, we observed a close association between both *PLA2G7* and *PTGS2* and the functionalities of mast cells and central memory CD4 T cells.

### 3.8. GSEA and GSVA of Two Hub Genes

In this study, we conducted an exploration of the functional roles of *PLA2G7* and *PTGS2* through GSEA. The low-expression cohorts of *PLA2G7* were found to be highly enriched in several pathways, including spliceosome, minoacyl-tRNA biosynthesis, olfactory transduction, and drug metabolism–cytochrome P450 pathways, as illustrated in Figure 7B. However, the low-expression cohorts of *PTGS2* exhibited significant enrichment in Toll-like receptor signaling, Fc gamma R-mediated phagocytosis, limonene and pinene degradation, maturity onset diabetes of the young (MODY), metabolism of xenobiotics by cytochrome P450, and drug metabolism–cytochrome P450 pathways, as demonstrated in Figure 7C. Notably, both *PLA2G7* and *PTGS2* were found to be associated with drug metabolism–cytochrome P450 pathways.

Gene set variation analysis (GSVA) was utilized to conduct a comprehensive investigation of the expression levels of these two hub genes in the pathological condition of PE. The results revealed that the downregulation of *PTGS2* expression was significantly associated with folate biosynthesis, dorsoventral axis formation, renal cell carcinoma, regulation of autophagy, glycosphingolipid biosynthesis (lacto and neolacto series), pantothenate and coenzyme A (CoA) biosynthesis, and O-glycan biosynthesis pathways. The downregulation of *PLA2G7* expression was significantly associated with aminoacyl tRNA biosynthesis as depicted in Figure 7D,E.

### 3.9. Drug–Gene Networks and Prediction of Key miRNAs and TFs

Potential drugs or molecular compounds that could reverse the downregulation of *PLA2G7* and *PTGS2* in PE were determined using DGIdb. As shown in the drug–gene interaction network (Figure 8A), darapladib and rilapladib were found to interact with *PLA2G7*. In addition, sedalate, valdecoxib, and carprofen were found to interact with *PTGS2*. Utilizing miRNet, we established miRNA and TF regulatory networks pertaining to *PLA2G7* and *PTGS2*. According to Figure 8B, ninety-one and twenty-three miRNAs were predicted to target *PTGS2* and *PLA2G7*, respectively. Specifically, both miR-335-5p and miR-124-3p had the potential to be related not only to *PTGS2*, but also to *PLA2G7*. Some target miRNAs of *PLA2G7* (e.g., miR-34a-5p, miR-126-3p, miR-27a-3p, miR-342-3p, miR-335-5p and miR-124-3p) and of *PTGS2* (e.g., miR-21-3p, mir-155-5p, mir146a-5p, miR-181a-5p, miR-335-5p and miR-124-3p) are involved in lipid metabolism and immunity. We found that 43 TFs, such as JUN, PPARG, FOS, STAT3, and RELA, not only regulate *PTGS2*, but also participate in regulating lipid metabolism and immunity. No TFs were found to regulate *PLA2G7*.

### 3.10. Validation of the mRNA Expression Levels of Hub Genes

Tissue samples were subjected to qPCR analysis to validate the expression levels of *PTGS2* and *PLA2G7*. Our findings from qRT–PCR analysis revealed that normal tissues exhibited significantly higher expression levels of *PTGS2* and *PLA2G7* than PE tissues. This observation suggests that these genes may function as novel suppressors in the context of PE (Figure 9A). These findings are consistent with the data obtained from the bioinformatics analysis (Appendix A).

## 4. Discussion

As a hypertensive disorder complicating pregnancy, PE induces fetal growth restriction, preterm birth, abortion, renal function damage, and other common complications [22]. It was reported that lipid metabolism, immune cell infiltration, oxidative stress, and inflammatory endothelial dysfunction played an important role in the pathogenesis of PE [22,26]. The objective of this study was to identify a pivotal LMRG-PE module to elucidate its pathogenesis and offer novel insights for future investigations.

This study employed publicly available databases to utilize single-cell RNA sequencing data to investigate the functional attributes associated with PE. Through dimensionality reduction and cell-type annotation, we identified five distinct cell types in PE. The majority of cells belonged to the monocyte and NK cell populations, suggesting potential involvement of monocyte and NK cells in the pathogenesis of PE. Additionally, we performed pseudotime analysis to visualize the clustering results of the cells along the trajectory of cellular differentiation. Then, we explored the relationship between PE and the immune system, highlighting the potential role of immune cells in reducing PE incidence. To identify markers associated with PE, we conducted differential gene expression analysis and subsequently constructed a weighted gene co-expression network using the WGCNA approach. The green co-expression module was chosen for analysis due to its highest correlation and strongest association with PE among the nine co-expression modules obtained. By combining DEG analysis and WGCNA, we identified common genes associated with PE, which were further narrowed down to 11 LMRG-PEs through intersection with literature-based marker-related genes. Enrichment analysis and PPI network analysis of these 11 genes revealed two significant hub genes, warranting further investigation. The reduced expression of placental lipases, including hormone-sensitive lipase, lipoprotein lipase, and endothelial lipase, may contribute to elevated levels of placental triglycerides in PE [6,41]. Previous studies have indicated that dyslipidemia can lead to elevated oxidative stress, endothelial dysfunction, and increased triglyceride levels as a consequence of diminished activity of lipoprotein lipases [42]. In PE, the occurrence of endothelial damage and lipid accumulation results in the development of atherosclerosis and abnormal narrowing of spiral arterioles [28].

After ROC curve analysis, we found that *PLA2G7* and *PTGS2* had high sensitivity and specificity in the diagnosis of PE. Thus, these two hub genes with decreased expression in the PE group might be potential biomarkers for predicting and diagnosing PE. Phospholipase A2 group VII (*PLA2G7*), also known as lipoprotein-associated phospholipase A2 (Lp-PLA2), is the gene encoding the protein platelet-activating factor acetylhydrolase (PAF-AH). This secreted enzyme plays a crucial role in the degradation of PAF into biologically inactive metabolites. Thromboxane production, platelet aggregation, and inflammation are mediated by the involvement of PAF and PAF-like oxidized phospholipids. The findings of this study suggest that the G994T variant of *PLA2G7* may be linked to reduced or absent activity of PAF-AH and altered distribution of enzymatic activity across lipoproteins. These alterations have the potential to promote heightened inflammation and oxidative stress, thereby increasing susceptibility to PE [43]. Prostaglandin-endoperoxide synthase 2 (*PTGS2*), also referred to as cyclooxygenase 2 (COX-2), functions as a pivotal enzyme in prostaglandin biosynthesis and mitogenesis. It has been postulated that COX-2 plays a regulatory role in the invasion of human trophoblast cells [44,45,46]. Multifactorial pregnancy disorders, including impaired placentation characterized by inadequate trophoblast invasion, have the potential to result in PE [47]. Yi et al. [48] discovered that the upregulation of COX-2 expression is facilitated by TGF-β1 through the activation of SMAD2/3-SMAD4 signaling. Moreover, the inhibition of trophoblast cell invasion by TGF-β1 necessitates the induction of COX-2. Prior studies [44] have demonstrated the ability of vitamin D to mitigate the risk of PE by downregulating both COX-2 expression and PGE2 signaling.

According to the GSEA results, the expression of *PLA2G7* exhibited significant associations with pathway enrichment in pathways related to spliceosomes [49], minoacyl-tRNA biosynthesis, olfactory transduction, and drug metabolism–cytochrome P450. In addition, the cohorts with low *PTGS2* expression demonstrated noteworthy enrichment in immune-related pathways [50], including Toll-like receptor signaling and Fc gamma R-mediated phagocytosis. In GSVA, the downregulation of *PTGS2* expression was significantly associated with various pathways [51], including folate biosynthesis, dorsoventral axis formation, renal cell carcinoma, regulation of autophagy, glycosphingolipid biosynthesis (lacto and neolacto series), pantothenate and CoA biosynthesis, and O-glycan biosynthesis. The downregulation of *PLA2G7* expression was significantly associated with aminoacyl tRNA biosynthesis [52].

However, there are certain limitations in our study that should be acknowledged. First, we relied exclusively on target data sourced from the GEO public database and employed biological algorithms for analysis, which may introduce inherent limitations and biases. Additionally, while we identified 11 hub genes as potential biomarkers associated with lipid metabolism in PE, it is crucial to conduct larger-scale, multicenter, prospective clinical cohort studies to evaluate the clinical applicability and validity of our findings. Our investigation specifically targeted protein-coding genes; however, emerging evidence suggests that noncoding RNAs, including long noncoding RNAs (lncRNAs) and microRNAs, exert significant influences on the pathogenesis of PE. Therefore, in future research, we will aim to address multiple aspects. First, our study has identified *PLA2G7* and *PTGS2* as potential diagnostic biomarkers; however, further validation through larger, multicenter, and prospective clinical cohort studies is necessary to establish the clinical applicability and reliability of our findings. Second, there is increasing evidence supporting the significant role of noncoding RNAs, such as long noncoding RNAs (lncRNAs) and microRNAs, in the pathogenesis of PE. Hence, our future research will investigate the involvement of these noncoding RNAs in the development of PE. Third, we plan to explore the role of the immune system in PE development, building on the potential involvement of immune cells, as indicated by our study. Understanding the interplay between immune system components and PE could enhance our comprehension of the pathogenesis of this disorder and may lead to the identification of new targeted therapies. Fourth, to gain further insights into the cellular and molecular mechanisms related to PE, it is essential to conduct additional investigations on the pathways identified by GSEA and GSVA to be associated with the expression of *PLA2G7* and *PTGS2*. By pursuing these future directions, our aim is to contribute to a more comprehensive understanding of PE, improve its diagnosis, and facilitate the development of potential therapeutics for this disorder.

## 5. Conclusions

In conclusion, our study utilizing single-cell analysis and WGCNA identified the potential involvement of *PLA2G7* and *PTGS2* in the progression of PE. The identification of these genes presents promising prospects as diagnostic biomarkers and therapeutic targets in the context of PE. To gain a comprehensive understanding of the functional mechanisms of these genes in the pathogenesis of PE, additional prospective and large-scale studies are imperative.

## Figures and Tables

**Figure 1 biomedicines-11-02328-f001:**
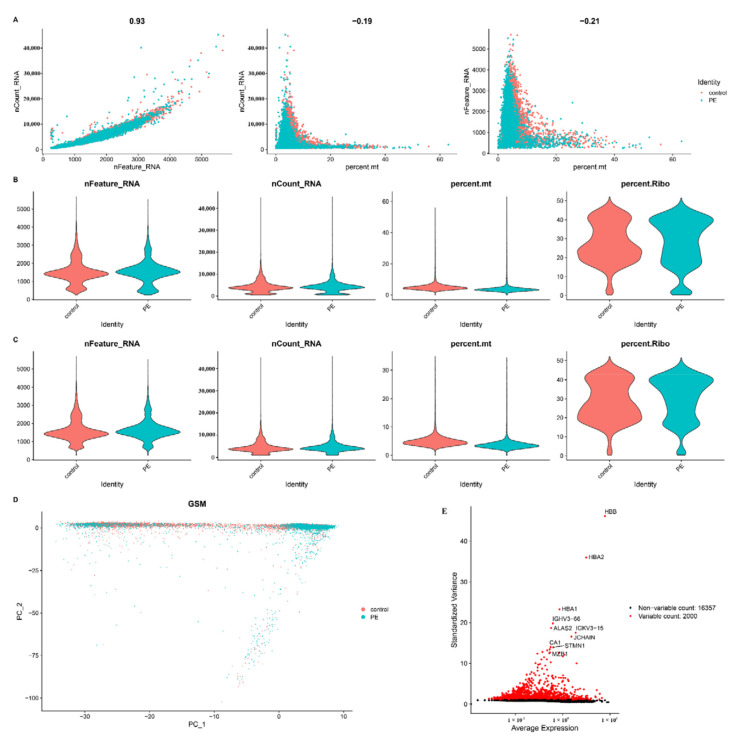
Single-cell sequencing analysis of PE and control samples. (**A**) Analysis of the correlation between nFeature and nCount, percent.mt and nCount, and percent.mt and nFeature. (**B**) Violin plots illustrating the RNA characteristic number (nFeature RNA) and absolute UMI count (nCount RNA) before quality control screening of cells. (**C**) The figure displays the total count number of each cell after quality analysis. (**D**) PCA plot. (**E**) Red dots represent the top 2000 high-variation genes obtained through variance analysis.

**Figure 2 biomedicines-11-02328-f002:**
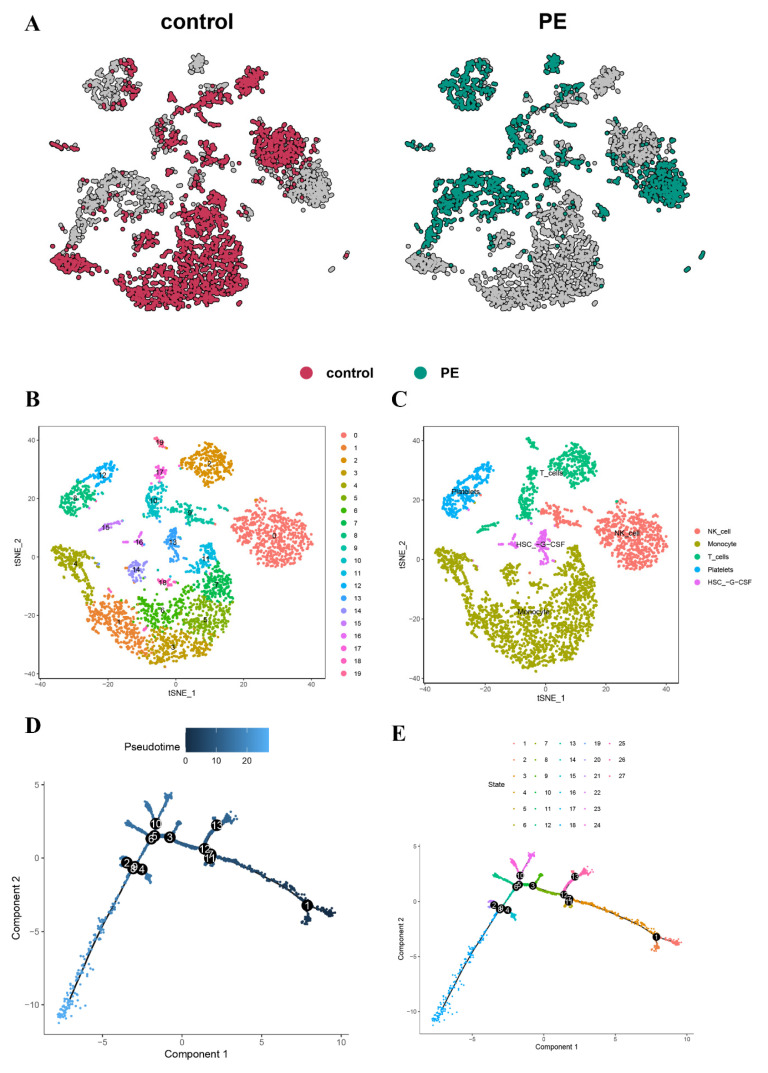
(**A**) The t-SNE plots represent tumor and normal cells in PE and normal samples, distinguished by different colors. (**B**) The t-SNE plot is colored according to various cell types. (**C**) Cell types are identified based on marker genes, enabling the reconstruction of the developmental relationship of trophoblast cells using pseudotime analysis. (**D**) The biaxial scatter plot visualizes the developmental trajectory of trophoblastic cells, with dark colors indicating early development. (**E**) The distribution of trophoblast states along the trajectory is illustrated.

**Figure 3 biomedicines-11-02328-f003:**
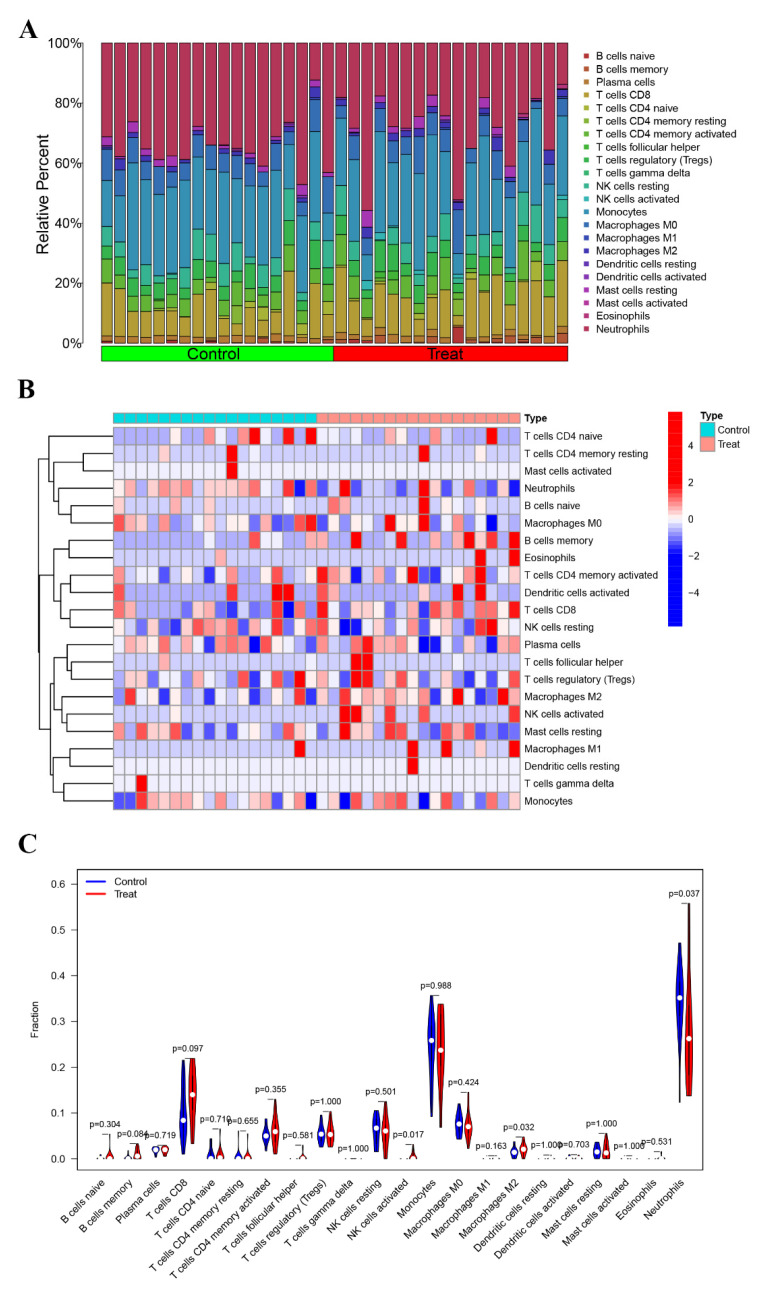
Immune infiltration analysis. (**A**) The bar plot visualizes the relative percentages of 22 immune cell types in each sample. Different colors represent distinct immune cell types. (**B**) Heatmap depicting the identification of differentially infiltrating immune cells. (**C**) The violin plot illustrates the differences in proportions of these immune cells between the control (Con) group and the PE group. The PE group is represented in red, and the Con group is represented in blue. A *p*-value < 0.05 indicates statistical significance.

**Figure 4 biomedicines-11-02328-f004:**
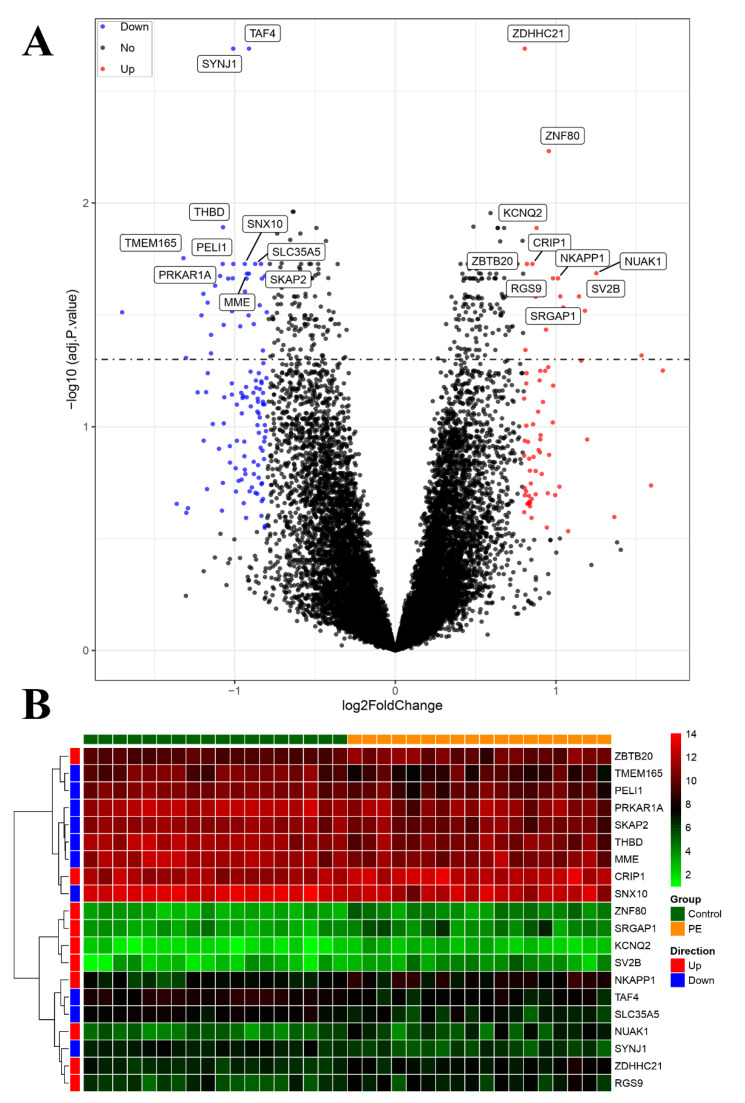
Visualization of differentially expressed genes (DEGs). (**A**) The volcano plot presents the DEGs identified based on the threshold (adjusted *p*-value < 0.05 and |logFC| > 1). (**B**) The heatmap displays the expression of the top 10 upregulated and downregulated genes, ordered by adjusted *p*-value.

**Figure 5 biomedicines-11-02328-f005:**
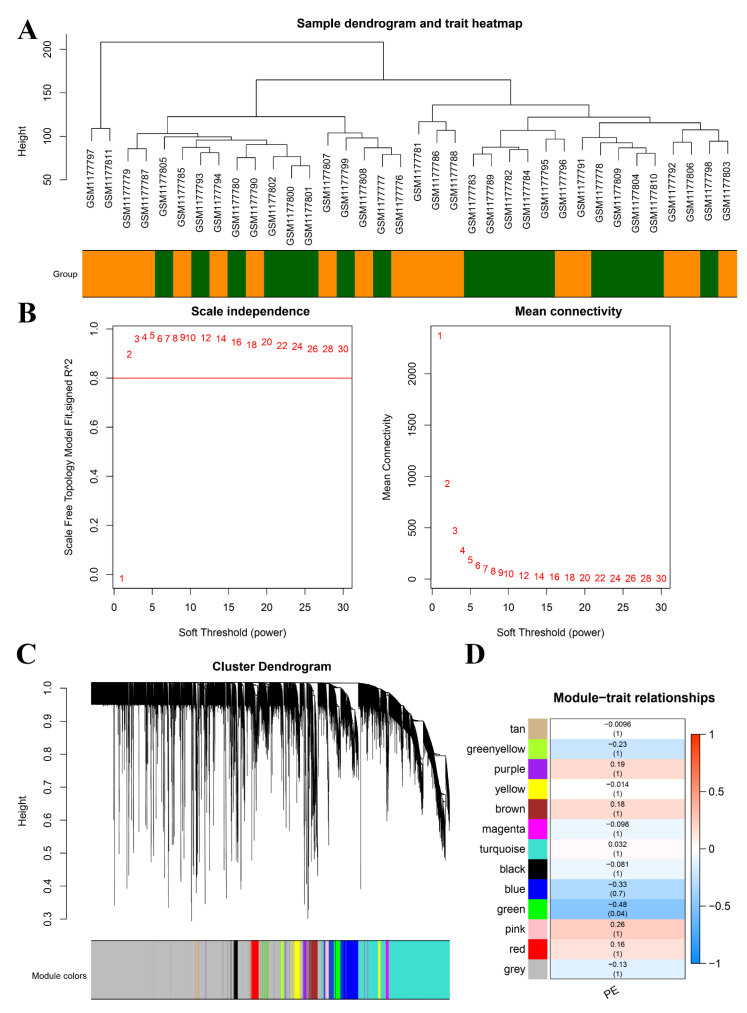
Construction of WGCNA co-expression modules and selection of hub modules. (**A**) Dendrogram of module eigengenes and heatmap of eigengene network. (**B**) Scale-free fit index for soft-thresholding powers. **Left**: relationship between soft-threshold and scale-free R2. **Right**: relationship between soft-threshold and mean connectivity. (**C**) Dendrogram of differentially expressed genes (DEGs) clustered in the training dataset. (**D**) Heatmap displaying the correlation between module eigengenes and pre-eclampsia (PE).

**Figure 6 biomedicines-11-02328-f006:**
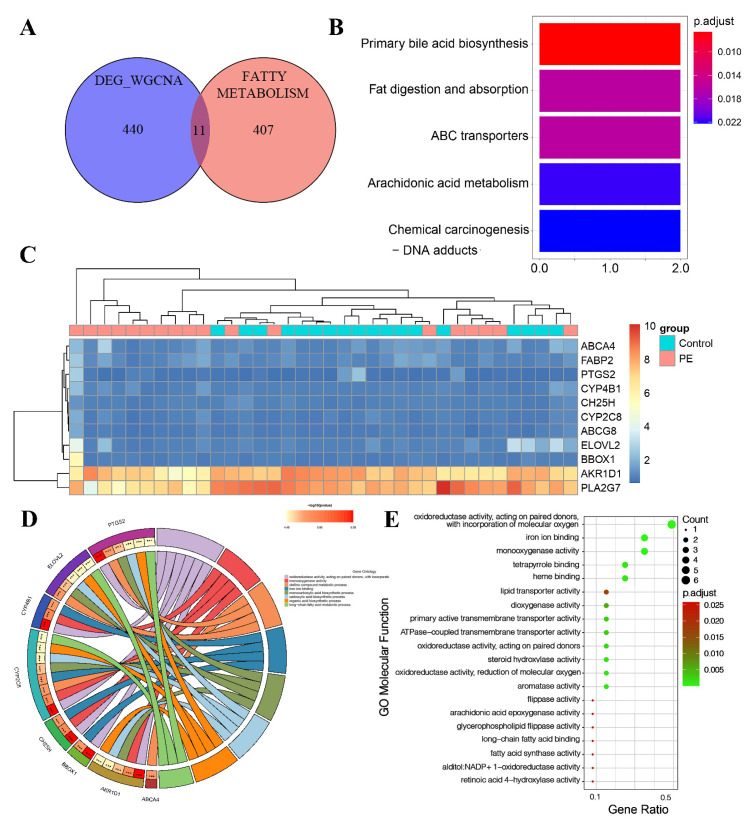
Screening of hub genes and candidate gene enrichment analysis. (**A**) Venn diagram illustrating the overlap of hub genes in the pre-eclampsia (PE)-related co-expression genes and LMRG (largest module of the weighted gene co-expression network analysis). (**B**) Heatmap displaying the expression of 11 candidate genes, ordered by adjusted *p*-value. (**C**) KEGG pathway analysis of the 11 candidate genes. (**D**) Gene Ontology enrichment analysis of the eleven hub genes in biological processes. (**E**) Gene Ontology enrichment analysis of the 11 hub genes in molecular functions. The color gradient from green to red represents the increasing significance of enrichment. The size of the dot represents the number of different genes included in the corresponding pathway.

**Figure 7 biomedicines-11-02328-f007:**
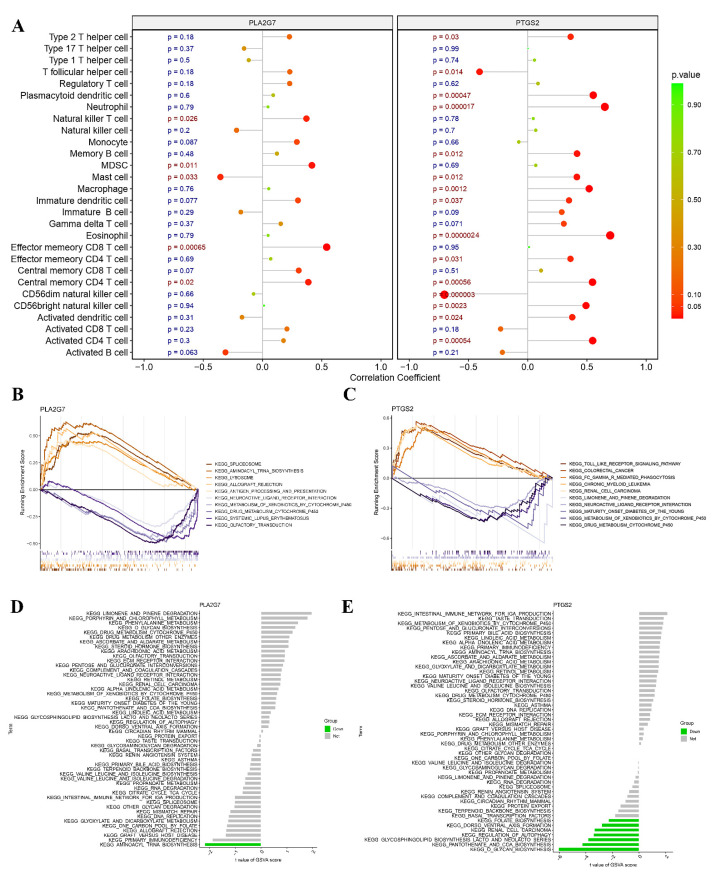
Relationship between the two hub genes and immune infiltration, gene set enrichment Analysis (GSEA), and targeted drugs. (**A**) Associations between immune cell infiltration and the two hub genes. (**B**,**C**) Top 5 pathways (based on GSEA enrichment score) enriched in the high-expression group and the low-expression group of *PLA2G7* (**B**) and *PTGS2* (**C**). (**D**,**E**) Gene set variation analysis (GSVA) of the two hub genes, *PLA2G7* and *PTGS2*, in pre-eclampsia (PE).

**Figure 8 biomedicines-11-02328-f008:**
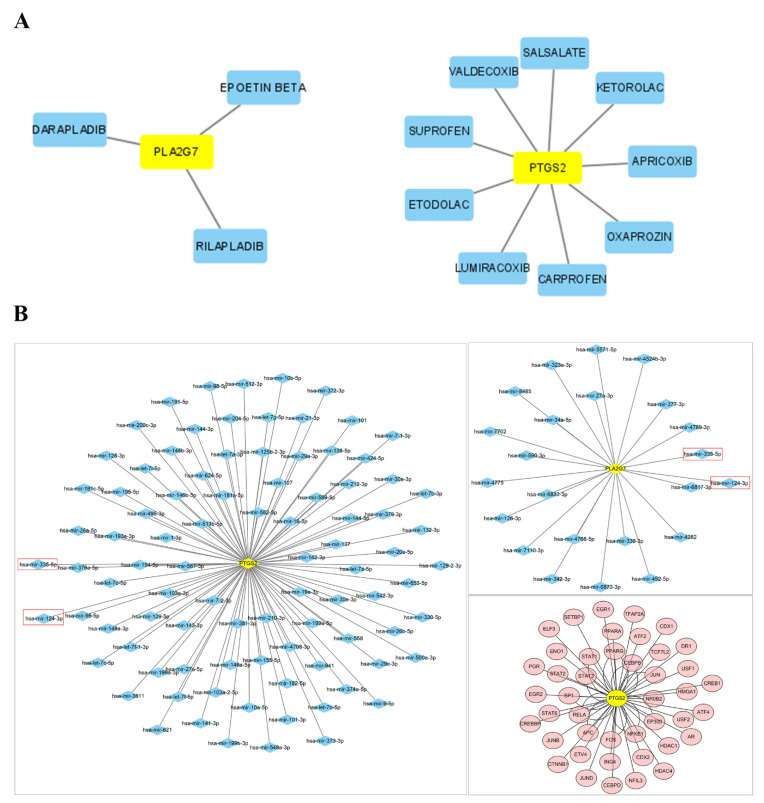
Network construction. (**A**) Drug–gene interaction network retrieved from the DGIdb database. Yellow nodes represent genes, and blue nodes represent drugs. (**B**) The networks of target gene–miRNA and target gene–TF. The red nodes are the TFs, the yellow nodes are the genes, and the blue nodes are the miRNAs. Red box highlights the emphasis on the potential association of miR-335-5p and miR-124-3p with both PTGS2 and PLA2G7.

**Figure 9 biomedicines-11-02328-f009:**
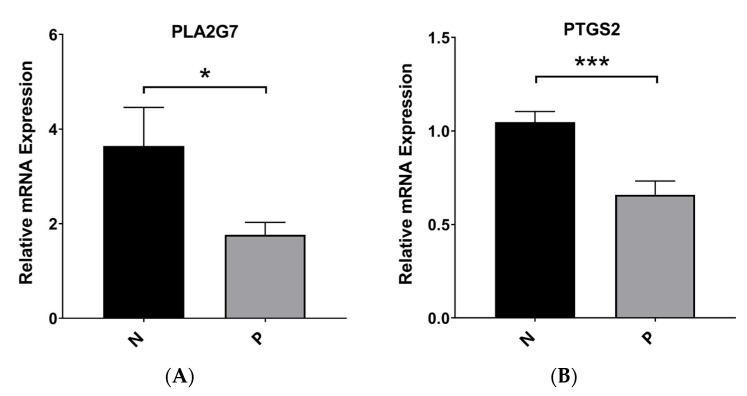
Robust experiment. (**A**) The mRNA expression of *PLA2G7* in PE and normal samples. (**B**) The mRNA expression of *PTGS2* in PE and normal samples. * *p* < 0.05 and *** *p* < 0.001.

## Data Availability

All data associated with this study are available in the main text or the Appendix A.

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
