# Peer review of "Single-Cell RNA Sequencing and Microarray Analysis Reveal the Role of Lipid-Metabolism-Related Genes and Cellular Immune Infiltration in Pre-Eclampsia and Identify Novel Biomarkers for Pre-Eclampsia"

_biomedicines, 2023, doi:10.3390/biomedicines11082328_

Round 1
Reviewer 1 Report
The manuscript entitled “Single-cell RNA sequencing and microarray analysis reveal the 2 role of lipid metabolism-related genes and cellular immune in- 3 filtration in pre-eclampsia and identify novel biomarkers for 4 pre-eclampsia.” The manuscript is significant but needs substantial revision for the abstract and conclusion section should be rewritten, the conclusion should have the limitations and recommendations and not only describe the results of the study. I can't find any emded figures within the manuscript , why and how???
Some major issues should be considered in the following points
- In the abstract : a lot of abbreviations were presented and with no full name description or list of abbreviation in the title page “ please, find how to solve?” it makes the reader so confused.
- What is the difference between “ "Seurat" R package and "Single R" package ? , what for?
- Line 41 and 42 , why the sentence is interrupted?
- Line 42 and 43 , “ Seizures, dyspnea, epigastric pain, and profound placental abruption are indictive symptoms of a critical terminal phase” This is not informative and needs proper citation.
- What are the pla- 44 cental or maternal-derived circulating biomarkers you mean ?
- Lines 46-48 : seems presented also with no citations.
- Line 55 : “ Non-steroidal anti-inflammatory drugs (NSAIDs) medications” why medications is added?
- “Since the syndrome of PE is hard to predict, we are committed to performing bioinformatics for” mention what bioinformatics do you mean?
- I found the aim of the work is not well presented and needs to be rewritten.
- I found the order of the presented sub-section in the method is confusing, while 2.7. subsection should be the first and reorder the others according to the order of methods.
- Mention in detail the process of placental tissue sample collection.
- Why you collected 16 sample exactly? I think it is very limited number to establish a study, relations and obvious conclusions.
- You mentioned that “ then fixed in 4% paraformaldehyde” why? If you did histopathological evaluation where is the complete methods and citations and presented figures ?????????/
- I can’t find any figures in the presented manuscript which are mentioned in all the results text , How? Even supplementary file or not , there is no any figures ,why?
- Agin you mention the objectives here in line 348 “ The objective of this study was to identify the pivotal LMRG-PE module in order to elucidate its pathogenesis and offer novel insights for future investigations ….???
-
Moderate language mistakes are available.
Author Response
Dear Reviewer:
We would like to thank Biomedicines for giving us the opportunity to revise our manuscript entitled “Single-cell RNA sequencing and microarray analysis reveal the role of lipid metabolism-related genes and cellular immune infiltration in pre-eclampsia and identify novel biomarkers for pre-eclampsia” (Manuscript ID: biomedicines-2511151), and the reviewers for their thoughtful comments on our previous manuscript. We have tried our best to revise the manuscript according to your kind and construction comments and suggestions. The ensuing content offers a thorough response to your comments and suggestions. We sincerely hope that the revisions articulated in our attached PDF can effectively address all your observations and recommendations.
Sincerely,
Dr. Cuifang Fan

Reviewer 2 Report
Poorly focussed technological 'fishing trip':
- It is unclear whether the authors focussed on acute atherosis and lipid metabolism, or on immune function in PE: this blurred focus is present from the Introduction to the Discussion. The finding of altered expression of PLA2G7 and PTGS2 was the caught fish.
- In the Introduction, the authors might describe the well known lesion of acute atherosis in pre-eclampsia. They are also ill-informed about NSAIDs: in most centers, NSAIDs are avoided in PE because of possible nephro-toxicity.
- Materials and methods are chaotic: should start with recruitment procedure, samples and population, sample preparation, use of technology, laboratory measurements, statistics.
Indeed, the recruitment of the PE and control population is unclear: purely convenience samples?
- Results focus on quantitative data: number of genes down and up, rather than a thoughtful flow of data acquisition.
- The Discussion recalls the fishing expedition, the two caught fish, and there we are. We would like to know what these genes mean for PE: any data on gene knock-out in general? previous data on PAF and cyclooxygenase?
Succinct, but often not to the point.
Author Response

(The authors gave the same response as above.)

Reviewer 3 Report
Dear Authors,
I am impressed by such sophisticated and genetically advanced study on preeclampsia pathophysiology and your simultaneous attempts to find a pharmacologic treatment. The research was based on placental tissue from PE and healthy patients from your hospital and genetic microarray-based RNA expression data obtained from a pubmed geo database.
In order to better understand your work I would suggest the following.
1/ In Introduction you should mention that we already have a combined screening test for PE and effective treatment as well:
Rolnik DL et al. Aspirin versus Placebo in Pregnancies at High Risk for Preterm Preeclampsia. N Engl J Med. 2017 Aug 17;377(7):613-622. doi: 10.1056/NEJMoa1704559. Epub 2017 Jun 28. PMID: 28657417.
2/ Make more clear your study design using a flowchart showing why and how you chose those genes and what was the rationale to select GSE48424.
3/ Present and refer to other studies using your methodology and genes selection process as it is hardly seen in the text.
4/ Line 150 says about Table 1 which is missing to evaluate the study group from your hospital, provide the flowchart of inclusion and exclusion criteria with lost cases if applicable.
5/ Discussion is missing the comparisons and references to other similar studies, reader does not know if anybody else did such research.
6/ Conclusions contain messages and suggestions about drugs which were not mentioned in the title nor were the aim of the study.
7/ Present your list of publications for each of the coauthors.
Thank you
Author Response

(The authors gave the same response as above.)

Reviewer 4 Report
I’ve read with attention the paper of Liu et al. that is potentially of interest. The background and aim of the study have been clearly defined. The methodology applied is overall correct, the results are reliable and adequately discussed. The choice of reference is adequate. I’ve only some minor comments:
- A graphical abstract and/or a resuming table should improve the readibility of the paper and the interest of the readers.
- All the final part of the paper is full of statements not supported by references. It shoud be improved.
- The authors should suggest the future direction/next step of their research
The quality of English language is quite good.
Author Response

(The authors gave the same response as above.)

Round 2
Reviewer 1 Report
The authors addressed most of required points while stilll no histopathological figures were presented and the manuscript needs English Editing.
The English needs Editing
Author Response
Dear Reviewer:
We thank Biomedicines for the opportunity to revise our manuscript, 'Single-cell RNA sequencing and microarray analysis reveal the role of lipid metabolism-related genes in pre-eclampsia' (Manuscript ID: biomedicines-2511151), and the reviewers for their insightful comments. We have carefully amended the manuscript based on your constructive suggestions. Enclosed in the attached PDF are detailed responses to your feedback. We hope that these revisions effectively address your observations and recommendations.
Sincerely,
Dr. Cuifang Fan

Reviewer 2 Report
Ok as it is.
The text, and especially the Discussion, remains "woody": "Other similar studies have been done by the following scholars before" (412) .... "Therefore, in future research we will have the following plans" (492)....
Author Response

(The authors gave the same response as above.)

Reviewer 3 Report
Dear Authors,
I appreciate your improvements following my suggestions. Now your manuscript appears much more comprehensive. Cosmetic correction (use capital letters) in Supplementary Figure 6: point 8. Heart Failure.
However, you still need to stress the presence of already effective screening in the first trimester by Fetal Medicine Foundation and by SFLT/PLGF ratio in 2nd and 3rd trimesters detecting women who will develop PE, mainly early-onset one (line 55-56).
Also aspirin was found to be effective in reducing early PE by almost 80% if initiated before 16 weeks 150mg until 36 wks.
That was a long awaited spectacular breakthrough in treating this dangerous condition so should be recalled and your study may give more insight in the pathogenesis thus better understanding of the PE origin.
Best wishes.
Author Response

(The authors gave the same response as above.)
